# Changing Attitudes toward the COVID-19 Vaccine among North Carolina Participants in the COVID-19 Community Research Partnership

**DOI:** 10.3390/vaccines9080916

**Published:** 2021-08-17

**Authors:** Chukwunyelu H. Enwezor, James E. Peacock, Austin L. Seals, Sharon L. Edelstein, Amy N. Hinkelman, Thomas F. Wierzba, Iqra Munawar, Patrick D. Maguire, William H. Lagarde, Michael S. Runyon, Michael A. Gibbs, Thomas R. Gallaher, John W. Sanders, David M. Herrington

**Affiliations:** 1Department of Internal Medicine, Section on Infectious Diseases, Wake Forest School of Medicine, Winston-Salem, NC 27517, USA; jpeacock@wakehealth.edu (J.E.P.J.); twierzba@wakehealth.edu (T.F.W.); imunawar@wakehealth.edu (I.M.); jwsander@wakehealth.edu (J.W.S.III); 2Department of Internal Medicine, Section on Cardiovascular Medicine, Wake Forest School of Medicine, Winston-Salem, NC 27517, USA; aseals@wakehealth.edu (A.L.S.); dherring@wakehealth.edu (D.M.H.); 3Biostatistics Center, George Washington University Milken School of Public Health, Washington, DC 20052, USA; sharone@bsc.gwu.edu; 4Jerry M. Wallace School of Osteopathic Medicine, Campbell University, Lillington, NC 27546, USA; hinkelman@campbell.edu; 5New Hanover Regional Medical Center, Wilmington, NC 28401, USA; patrick.maguire@nhrmc.org; 6Wake Med Health and Hospitals, Raleigh, NC 27610, USA; blagarde@wakemed.org; 7Atrium Health, Charlotte, NC 28204, USA; michael.runyon@atriumhealth.org (M.S.R.); michael.gibbs@atriumhealth.org (M.A.G.); 8Vidant Health, Greenville, NC 27834, USA; thomas.gallaher@vidanthealth.com

**Keywords:** vaccine hesitancy, vaccine intent, COVID-19, survey, North Carolina, vaccination, vaccine uptake, demographic groups

## Abstract

Coronavirus Disease-2019 (COVID-19) vaccine acceptance is variable. We surveyed participants in the COVID-19 Community Research Partnership from 17 December 2020 to 13 January 2021 to assess vaccine receptiveness. Vaccine uptake was then monitored until 15 May 2021; 20,232 participants responded to the receptiveness survey with vaccination status accessed in 18,874 participants via daily follow-up surveys (participants not completing daily surveys ≥30 days to 15 May 2021, were excluded). In the initial survey, 4802 (23.8%) were vaccine hesitant. Hesitancy was most apparent in women (Adjusted RR 0.93, *p* < 0.001), Black Americans (Adjusted RR 1.39, 1.41, 1.31 to non-Hispanic Whites, Other, and Hispanic or Latino, respectively *p* < 0.001), healthcare workers (Adjusted RR 0.93, *p* < 0.001), suburbanites (ref. Urban Adjusted RR 0.85, 0.90 to urban and rural dwellers, respectively, *p* < 0.01), and those previously diagnosed with COVID-19 (RR 1.20, *p* < 0.01). Those <50 years were also less accepting of vaccination. Subsequent vaccine uptake was 99% in non-hesitant participants. For those who were unsure, preferred not to answer, or answered “no”, vaccination rates were 80% (Adjusted RR 0.86, *p* < 0.0001), 78% (Adjusted RR 0.83, *p* < 0.0001), and 52.7% (Adjusted RR 0.65, *p* < 0.0001), respectively. These findings suggest that initial intent did not correlate with vaccine uptake in our cohort.

## 1. Introduction

Three COVID-19 vaccines have been approved for Emergency Use Authorization (EUA) in the United States by the Food and Drug Administration (FDA) [1] Vaccine acceptance and willingness to undergo vaccination are not universal with some national surveys suggesting that only 50–55% of respondents would be willing to receive the COVID-19 vaccine [2,3]. In North Carolina, early polls revealed intent at only 40–45% [4].

To assess the extent of vaccine hesitancy in North Carolina, we conducted a survey between 17 December 2020, and 13 January 2021, of 20,232 individuals affiliated with five medical centers from differing geographic regions of North Carolina. This was followed by a daily survey of respondents from the same cohort to determine subsequent vaccine uptake rates. This report presents the key findings from those two surveys.

## 2. Materials and Methods

The COVID-19 Community Research Partnership (CCRP) is a multi-site, prospective study combining daily electronic symptom surveillance, longitudinal serologic surveillance, and electronic health record capture [5]. Demographic and survey data are collected via a secure, HIPAA-compliant, online portal. The study has received approval by a centralized Institutional Review Board (Wake Forest Baptist Health). Five of the 10 CCRP sites participated in this sub-study. Those sites were Campbell University in Buies Creek, NC, New Hanover Regional Medical Center in Wilmington, NC, Wake Forest Baptist Health in Winston-Salem, NC, WakeMed Health and Hospitals in Raleigh, NC, and Vidant Health, in Greenville, NC.

Basic demographic data captured for all CCRP participants included age, sex, previous COVID-19 diagnosis status, community of residence, and race/ethnicity. Counties of residence were characterized as urban, suburban, or rural utilizing the North Carolina rural center counties map. Densities were calculated based on the 2014 census population estimates [6] Participants were also classified as to whether they were healthcare workers with all healthcare-related vocations and disciplines included in that category. In addition to the daily CCRP surveys where participants are asked questions about COVID-like symptoms, infections, and their vaccination status, they were asked to complete a single multiple-choice mini-survey on attitudes about COVID-19 vaccination. Participants completed these surveys using an online Patient Monitoring System application developed by Oracle Corporation (Redwood, CA, USA). There were four choices for vaccine intent: yes, no, undecided, and prefer not to answer. For participants who did not respond “yes” to the vaccine intent question, a follow-up question asked participants to specify reasons for vaccine hesitancy (Figure 1). The vaccine attitudes survey data was collected between 17 December 2020 and 13 January 2021.

To correlate the association of demographic characteristics with vaccine intent, binomial regression was implemented. Vaccine intent responses were categorized into two groups for this analysis: Yes and No/Undecided. To assess how demographic characteristics were associated with vaccine receptiveness, multivariate binomial regression was used with a log link function. The resulting coefficient estimates were exponentiated (eb) to calculate relative risk. The relative risk of responding ‘Yes’ was calculated for each demographic variable, along with 95% confidence intervals. Previous COVID-19 Diagnosis was self-reported information included as a part of the enrollment questionnaire (Appendix A). The impact of a prior diagnosis of COVID-19 on vaccine receptiveness was also analyzed in a similar fashion. *p*-values < 0.001 were considered significant.

In addition to the vaccine attitudes survey, daily CCRP survey data through 15 May 2021, was used to assess which participants reported receiving at least one COVID-19 vaccine dose. The vaccine rate was stratified by the vaccine intent and participants’ characteristics (Table 1 and Table 2). Unvaccinated participants who did not complete the daily CCRP survey 30 days or more prior to 15 May 2021 were excluded from the vaccine rate calculation. All statistical analysis was performed using SAS 9.4 (SAS Institute, Cary, NC, USA).

## 3. Results

A total of 20,232 people completed the initial mini-survey (Campbell University, n = 147; New Hanover Regional Medical Center, n = 641; Wake Forest Baptist Health, n = 16,058; Wake-Med Health and Hospitals, n = 2419; and Vidant Health, n = 967). The follow-up survey was completed by 18,874 respondents of the same cohort. This decrease in number of participants was due to exclusion of unvaccinated individuals who did not complete the daily follow-up survey 30 days or more prior to 15 May 2021.

In the initial survey, 76.2% of participants indicated their intent to get the vaccine. Participants were grouped by race, gender, age, residence locale, healthcare worker status, and previous COVID-19 infection (Table 3). Comparative analysis showed that the most likely respondents to get the vaccine were non-Hispanic Whites, males, urban dwellers, and non-healthcare workers (Table 4). Respondents with no previous COVID-19 diagnosis were also more likely to accept the vaccine. Intent to get the vaccine increased with age, with those >70 years most likely to get the vaccine. A total of 4810 (23.8%) participants in the initial survey did not answer “yes” to getting the vaccine. Black Americans were the least likely to show intent to get the vaccine. By residence locale, suburban residents were the least likely to get the vaccine and by age, those aged 40–49 showed the least intent to get vaccinated. The most common reasons for not accepting the vaccine, were concerns about safety (67%) and efficacy (15%) (Figure 1).

The follow-up survey revealed some interesting data about vaccine uptake in these groups. Overall, of the 18,874 individuals that participated in the daily survey until 15 May 2021, 17,461 (92.5%) obtained a COVID-19 vaccine. Among the participants that had intent to get the vaccine, there was a 98.5% vaccine uptake. Of those who did not answer “yes” to the initial survey, there was a 70% vaccine uptake (Table 1). Within this group, there was an 80% (Adjusted RR 0.86, *p* < 0.0001) vaccine uptake among those that were undecided, 77.6% (Adjusted RR 0.83, *p* < 0.0001) uptake among those that preferred not to answer, and a 52.7% (Adjusted RR 0.65, *p* < 0.0001) uptake among those that answered “no” to getting the vaccine in the initial survey.

Interestingly, Black Americans who were the least likely to accept the vaccine based on expressed pre-vaccination attitudes had an 88.9% vaccine uptake (Table 2). There was no difference in vaccination between healthcare workers and non-healthcare workers. Reported vaccination was >89% in all age ranges, with those >70-years with the highest vaccine uptake (97.2%). Males had a higher uptake than females (94% v 91.8%). Suburban dwellers, who were least likely to indicate vaccine acceptance in the initial survey, had an 88.4% uptake. Those respondents who reported no previous COVID-19 diagnosis were more likely to be vaccinated than to those with a previous COVID-19 diagnosis (92.7% v 81.9%). Despite this difference, uptake in those with a previous COVID-19 diagnosis was higher than expected given that only 60.9% had intent to get the vaccine.

## 4. Discussion

Multiple surveys have looked at COVID-19 vaccine hesitancy [2,3,7,8,9,10] Surveys conducted before and in the early phases of the vaccine rollout showed vaccine hesitancy of 50–55%. Black, indigenous, and people of color (BIPOC) communities, which are disproportionately affected by the pandemic, had the highest level of hesitancy [11,12,13,14]. Recent data have shown a change in attitudes in BIPOC communities [15,16]. This is a trend that seems to be the same across all categories represented in our survey. What is especially noteworthy is that vaccine uptake is quite high in those participants that were undecided or preferred not to answer. This observation emphasizes the dynamic nature of vaccine hesitancy. Vaccine hesitancy exists along a spectrum with the extremes of the spectrum being those who completely refuse and those who completely accept vaccines [17]. Individuals within this spectrum are not static and may shift across the spectrum over time, depending upon the current social and clinical context and available vaccine options. In the setting of the COVID-19 pandemic, time is of the essence because the faster we get to herd immunity, the better our chances of avoiding emergence of virulent and evasive variants [18]. Increasing uptake in those in the middle of the spectrum (refuse but unsure, undecided, accept but unsure) may help us achieve herd immunity. Although we did not look at what led to changes in attitudes, it is likely that public health educational interventions which increased confidence in the safety and efficacy of the COVID-19 vaccine may have led individuals in the continuum of vaccine hesitancy to shift towards acceptance [19,20,21]. A July 2021, vaccine monitor report by the Kaiser Family Foundation found that previously hesitant individuals changed their minds because of influences from family, friends, and personal doctors [22]. As we attempt to reach herd immunity, it is important that we recognize what impact these interventions have and implement them in communities with high levels of vaccine hesitancy and low uptake.

There are several potential limitations of this study. First, our data represent responses from North Carolinians enrolled into an ongoing research study through regional healthcare systems and may not be representative of national data. Our study volunteers are likely to be better connected to a healthcare system and more comfortable with electronic communication than the general community. Second, this survey provides only a snapshot in time and, as demonstrated by out follow-up survey, attitudes do change over time. Third, the demographics of our survey participants may not reflect national demographics. Fourth, our finding of more hesitancy among suburban residents may simply reflect limitations in our method for characterizing counties as it is difficult to accurately describe many counties in North Carolina as entirely urban, suburban, or rural. Fifth, we did not detail the reasons why our participants had a change in attitude trending towards increased acceptance. Finally, survey results might not be comparable to other national or state polls or surveys due to potential differences in survey methods, sample population, and questions related to willingness to get the vaccine.

In conclusion, this survey examined attitudes towards the COVID-19 vaccine in the initial phases of the vaccine rollout, and then followed the same cohort over subsequent months looking at uptake. Initial intent did not correlate with uptake in our North Carolina cohort. Vaccine uptake in participants that did not answer “yes” to the initial survey was 70% which offers some insight that broader acceptance of vaccination may continue to evolve over time.

## Figures and Tables

**Figure 1 vaccines-09-00916-f001:**
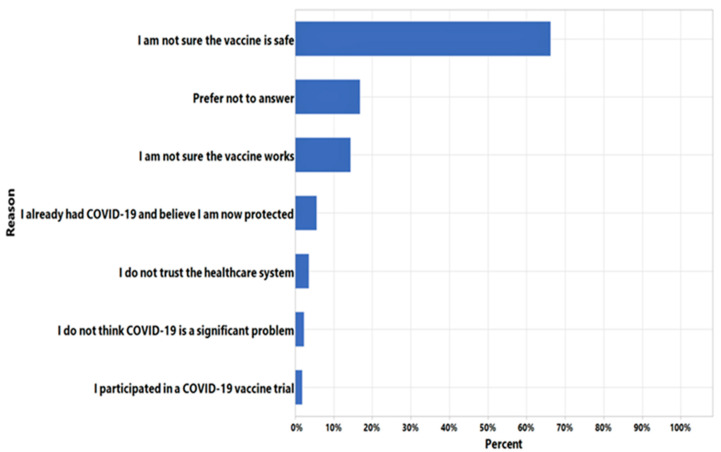
Reasons given by participants who did not respond “yes” on the survey.

**Table 1 vaccines-09-00916-t001:** Vaccine intent and vaccination status with Relative Risk of being vaccinated. n—number, CI—confidence interval, ref—reference.

	Total	Vaccine Status						
		Vaccinated		Non-Vaccinated				
	n	n	Percent	n	Percent	Relative Risk	95% CI	*p*-Value
TOTAL	18,874	17,461	92.5	1413	7.5			
Vaccine Intent								
Yes	14,810	14,582	98.5	228	1.5	ref.	ref.	ref.
No	1358	715	52.7	643	47.3	0.65	0.62–0.67	<0.0001
Undecided	2403	1929	80.3	474	19.7	0.86	0.84–0.87	<0.0001
Prefer no to answer	303	235	77.6	68	1.6	0.83	0.79–0.88	<0.0001

**Table 2 vaccines-09-00916-t002:** Vaccination status by participant characteristics.

	Total	Vaccine Status			
		Vaccinated		Non-Vaccinated	
	n	n	Percent	n	Percent
TOTAL	18,874	17,462	92.5	1413	7.5
SEX					
Women	12,835	11,780	91.8	1055	8.2
Men	6039	5681	94.1	358	5.9
AGE GROUP					
<30	895	804	89.8	91	10.2
30–39	2766	2483	89.8	283	10.2
40–49	3577	3197	89.4	380	10.6
50–59	4001	3636	90.9	365	9.1
60–69	4715	4503	95.5	212	4.5
70+	2920	2838	97.2	82	2.8
ETHNICITY					
Black or African American	811	721	88.9	90	11.1
Hispanic or Latino	379	350	92.3	29	7.7
Other	692	643	92.9	49	7.1
White (not Hispanic/Latino)	16,992	15,747	92.7	1245	7.3
HEALTHCARE WORKER					
N	13,997	12,956	92.6	1041	7.4
Y	4877	4505	92.4	372	7.6
PREVIOUS COVID-19 DIAGNOSIS					
N	18,482	17,140	92.7	1342	7.3
Y	392	321	81.9	71	18.1
SITE					
Campbell University	134	123	91.8	11	8.2
New Hanover Regional Medical Center	612	576	94.1	36	5.9
Vidant Health	873	804	92.1	69	7.9
Wake Forest Baptist Health	15,022	13,835	92.1	1187	7.9
WakeMed Health and Hospitals	2233	2123	95.1	110	4.9
COUNTY CLASS					
Rural	6118	5521	90.2	597	9.8
Suburban	1533	1355	88.4	178	11.6
Urban	11,223	10,585	94.3	638	5.7

**Table 3 vaccines-09-00916-t003:** Vaccine intent responses grouped by sex, age, race & ethnicity, healthcare worker status, previous COVID-19 diagnosis, Site and county class. n—no; P—prefer not to answer; N—no; Y—yes.

	Total	Vaccine Intent			
		No/Undecided/Prefer Not to Answer		Yes	
	n	N	Percent	n	Percent
TOTAL	20,232	4810	23.8	15,422	76.2
SEX					
Women	13,784	3688	26.8	10,096	73.2
Men	6448	1122	17.4	5326	82.6
AGE GROUP					
<30	1096	322	29.4	774	70.6
30–39	3086	889	28.8	2197	71.2
40–49	3897	1175	30.2	2722	69.8
50–59	4256	1180	27.7	3076	72.3
60–69	4898	865	17.7	4033	82.3
70+	2999	379	12.6	2620	87.4
ETHNICITY/RACE					
Black or African American	932	426	45.7	506	54.3
Hispanic or Latino	413	122	29.5	291	70.5
Other	758	179	23.6	579	76.4
White (not Hispanic/Latino)	18,129	4083	22.5	14,046	77.5
HEALTHCARE WORKER					
N	15,062	3466	23	11,596	77
Y	5170	1344	26	3826	74
PREVIOUS COVID-19 DIAGNOSIS					
N	19,756	4624	23.4	15,132	76.6
Y	476	186	39.1	290	60.9
SITE					
Campbell University	147	48	32.7	99	67.3
New Hanover Regional Medical Center	641	100	15.6	541	84.4
Vidant Health	967	285	29.5	682	70.5
Wake Forest Baptist Health	16,058	3917	24.4	12,141	75.6
WakeMed Health and Hospitals	2419	460	19	1959	81
COUNTY CLASS					
Rural	6645	1916	28.8	4729	71.2
Suburban	1669	530	31.8	1139	68.2
Urban	11,918	2364	19.8	9554	80.2

**Table 4 vaccines-09-00916-t004:** Vaccine acceptance multivariate analysis grouped by sex, age, race & ethnicity, healthcare worker status, previous COVID-19 diagnosis, Site and county class. CI—Confidence interval; ChiSq—Chi square. Ref.—reference group.

		Vaccine Acceptance Multivariate Model Estimates			
Parameter		Estimate	Adj. Relative Risk	Relative Risk 95% CI	Pr > ChiSq
SEX	Women	−0.07	0.93	0.92–0.94	<0.0001
	Men	ref.	ref.	ref.	
AGE GROUP	<30	−0.17	0.85	0.81–0.88	<0.0001
	30–39	−0.16	0.85	0.83–0.87	<0.0001
	40–49	−0.17	0.84	0.82–0.86	<0.0001
	50–59	−0.15	0.86	0.84–0.88	<0.0001
	60–69	−0.03	0.97	0.95–0.98	0.0001
	≥70	ref.	ref.	ref.	
ETHNICITY/RACE	Hispanic or Latino	0.27	1.31	1.20–1.42	<0.0001
	Other	0.34	1.41	1.32–1.51	<0.0001
	White (not Hispanic/Latino)	0.33	1.39	1.31–1.48	<0.0001
	Black or African American	ref.	ref.	ref.	
HEALTHCARE WORKER	No	−0.03	0.97	0.96–0.99	0.0014
	Yes	ref.	ref.	ref.	
PREVIOUS COVID-19 DIAGNOSIS	No	0.18	1.20	1.11–1.28	<0.0001
	Yes	ref.	ref.	ref.	
COUNTY CLASS	Rural	−0.10	0.90	0.89–0.92	<0.0001
	Suburban	−0.16	0.85	0.83–0.88	<0.0001
	Urban	ref.	ref.	ref.	

## Data Availability

The datasets used and/or analyzed during the current study are available from the corresponding author upon request.

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
