# Peer review of "Changing Attitudes toward the COVID-19 Vaccine among North Carolina Participants in the COVID-19 Community Research Partnership"

_vaccines, 2021, doi:10.3390/vaccines9080916_

Round 1
Reviewer 1 Report
Chukwunyelu H Enwezor et al. conducted a brief survey between December 17, 2020, and January 13, 2021, about the acceptance level of vaccination in CCRP affiliated with five medical centers in North Carolina. From the results, they found that 70+ persons and respondents with no previous COVID-19 diagnosis were most willing to be vaccinated compared to other groups. women, African Americans, and suburban participants were less willing to get a COVID-19 vaccine. 82% of unwilling people indicated that the safety and efficacy of the vaccine were their main worries.
The manuscript is somewhat wordy and some expressions do not convey the idea, for example, “Our survey results suggest that vaccination hesitancy may be a potential impediment to a successful vaccination campaign” it is common sense that hesitancy is a potential impediment to a successful vaccination campaign, not from your results.
The manuscript is outdated and this survey only provided a short time during the early rollout of the vaccines, and attitudes have changed a lot with more nationwide and local interventions to eliminate the hesitancy. The survey lacks scientific significance and just a kind of record and does not suit to publish in Vaccines.
Author Response
Dear Reviewer,
Thank you for your comments. Our goal has always been to communicate the results of our survey the best way we can.
We agree that the data was dated. This is why we have added follow-up data to our updated manuscript. We followed our cohort to see if their initial attitudes towards the vaccines correlate with uptake. I think the results importantly convey changing attitudes towards acceptance even in initially hesitant participants. We continue to follow the same cohort of >20,000 participants and it is likely that we would continue to get updated data as attitudes change.
I hope this version is more concise and less wordy.
Looking forward to your feedback.
Thanks,
Chukwunyelu Enwezor, MD

Reviewer 2 Report
Dear Editors,
The part of this study concerning health workers is of interest to the whole scientific community since it highlights that the concerns about vaccines in general is deeper than if it was only touching the non scientific population.
Please see my recommendations below.
Best
This study proposed by Enwezor et al reports how vaccination against COViD-19 was accepted among North Carolina participants between December 2020 and January 2021. The subject is quite interesting since it also includes the attitude of healthcare workers. While such studies always have bias, they are quite well highlighted here even if a little bit late, and it reveals that the attitude towards vaccines may be more societal than based on the knowledge since it showed that even healthcare workers were not that likely to accept vaccination compared to others. The manuscript could be better organized and few additional elements would improve it.
1-The limitations of the study are pretty well presented but only in the last paragraph just before the conclusion. The manuscript would read better if it was at the beginning of the discussion since the reader think about all these biases from the beginning.
2-Table 2 is not easy to read compared to table 1, thus it is unclear what does it really add to the story since it is poorly described or used.
3-still in table 2: it is unclear what means “Ref” which is repeated multiple times. Does it mean it was published elsewhere separately? Where? Men were not included? It is really unclear.
4-Data about healthcare worker would merit a little bit more details. For example among health workers less likely to accept vaccine what was the percentage of nurse, medical doctors, surgeons etc? This could improve the story
5-In the introduction, multiple message of error are present showing a problem in references “Error! Reference source not found”. Please correct
Author Response
Dear Reviewer,
Thank you for your comments. Our goal has always been to communicate the results of our survey the best way we can.
In response to your comments as listed:
- We really want to make the key points in the discussion before moving on to the limitations which are many as expected in a survey
- There have been changes made to the tables
- Reference is in relation to the variable that was used as a reference. For instance, for race and ethnicity, the data for Black Americans was used as a reference to other groups in that category
- Our survey does not subcategorize healthcare workers. We mentioned it in the methods. The survey was not focused primarily on healthcare workers. We may consider this as a standalone survey in the future given the wide array of healthcare workers that exist.
- We have corrected these references.
Our survey has been updated with more recent data and reads differently. Looking forward to your feedback.
Best Regards,
Chukwunyelu Enwezor MD

Reviewer 3 Report
In the article, a survey about the attitudes toward COVID-19 vaccine among NC residents has been done. The results show the vaccinated population was more than these who were willing to accept vaccination, and suggest that initial intent did not correlate with actual vaccination. The finding is helpful for public health workers and governments to promote vaccination in communities. A few modifications are needed to improve the data presentation.
- Tab 4 is basically a repeat of tab 3. Please merge the two tabs as one. The title of tab 4 is not corresponding to the content of the table. There is no information about sex, age, race/ethnicity, professional, diagnosis, and county class there.
- Tab 2 is confused. Why are there two columns of data for "relative risk 95% CI"? The text of the first row (Vaccine acceptance multivariate model estimates) is not properly located.
Author Response
Dear reviewer,
Thank you for your comments and suggestions.
We have made modifications to the tables.
We deleted table 3 and left table 4. Made the needed edits to table 2.
Thank you reviewing our paper and helping us make it better.
Sincerely,
Chuka Enwezor MD

Reviewer 4 Report
General Impression
The authors present a report on the connection between COVID19 vaccine hesitancy and the actual likelihood to get vaccinated. The data suggest that the majority of individuals who identified as either vaccine hesitant or undecided by January 13th, 2021 ended up being vaccinated by May 12th. Data on vaccination outcomes are broken down by population subgroups and analyzed appropriately by multivariate binomial regression. While experimental design, data analysis and discussion are appropriate, this reviewer’s enthusiasm is dampened by the purely descriptive nature of the study. While the data clearly show that individuals who had reservations about vaccination had a change of mind during the 5-month observation period, no data have been collected to explain this phenomenon and the discussion thus remains speculative. Meanwhile, the Kaiser Family Foundation published a detailed report on the factors that lead vaccine hesitant individuals to change their minds (KFF COVID-19 Vaccine Monitor). The Kaiser findings need to be discussed. The unique aspects of the study methodology and the technical soundness of the analysis provide some valuable information and might publication.
Specific points
Material and methods – please specify the means by which survey data were collected. If survey participation requires computer literacy or ownership of a late-model cellphone, this confounding factor also needs to be discussed.
Discussion – the first sentence quotes reference 10 twice.
Author Response
Hello,
Thank you for reviewing the paper. As you have noted, our survey is mostly descriptive. Our goal with this survey was primarily to describe changing attitudes in a large cohort of NC residents. As this is an ongoing study of growing participants, we may be able to reference the reasons for changing attitudes hopefully, as we modify our questionnaire in future papers. We did mention our inability to detail the reasons for changing attitudes as a limitation.
I did add the recent KFF study to our discussion briefly. That survey was published after we submitted our paper in May. We think our manuscript could be used, with other surveys like the KFF survey, as a reference for public health individuals that want to buttress the point that attitudes can change towards vaccine acceptance even among hard "no" individuals.
The means by which survey data was collected has been added to the materials and methods. Selection of individuals with more comfort with electronic communication was mentioned as a limitation to the study.
Corrected double referencing.
Thank you for taking the time to review our manuscript and helping us make it better.
Sincerely,
Chukwunyelu Enwezor MD

Round 2
Reviewer 3 Report
The defects in the tables have been corrected.
Reviewer 4 Report
Thank you for addressing my points of criticism and for the clarifications. I find the manuscript improved and support publication.